# Kinetic Studies on Radical Scavenging Activity of Kaempferol Decreased by Sn(II) Binding

**DOI:** 10.3390/molecules25081975

**Published:** 2020-04-23

**Authors:** Zhi-Yin Yang, Ling-Ling Qian, Yi Xu, Meng-Ting Song, Chao Liu, Rui-Min Han, Jian-Ping Zhang, Leif H. Skibsted

**Affiliations:** 1Department of chemistry, Renmin University of China, No. 59 ZhongGuanCun Street, Beijing 100872, China; 2016201111@ruc.edu.cn (Z.-Y.Y.); qianlingling@ruc.edu.cn (L.-L.Q.); yixu@ruc.edu.cn (Y.X.); mtsong2019@ruc.edu.cn (M.-T.S.); 2018102313@ruc.edu.cn (C.L.); jpzhang@ruc.edu.cn (J.-P.Z.); 2Department of Food Science, University of Copenhagen, Rolighedsvej 26, DK-1958 Frederiksberg C, Denmark; ls@food.ku.dk

**Keywords:** metal-flavonoid complex, radical scavenging, Sn(II), kaempferol

## Abstract

Sn(II) binds to kaempferol (HKaem, 3,4′,5,7-tetrahydroxy-2-(4-hydroxyphenyl)-4*H*-1-benzopyran-4-one) at the 3,4-site forming [Sn(II)(Kaem)_2_] complex in ethanol. DPPH^•^ scavenging efficiency of HKaem is dramatically decreased by SnCl_2_ coordination due to formation of acid inhibiting deprotonation of HKaem as ligands and thus reduces the radical scavenging activity of the complex via a sequential proton-loss electron transfer (SPLET) mechanism. Moderate decreases in the radical scavenging of HKaem are observed by Sn(CH_3_COO)_2_ coordination and by contact between Sn and HKaem, in agreement with the increase in the oxidation potential of the complex compared to HKaem, leading to a decrease in antioxidant efficiency for fruits and vegetables with Sn as package materials.

## 1. Introduction

Flavonoids are a group of natural products with a variety of biological activities such as antioxidant, anti-inflammatory and anti-cancer [1,2,3,4,5,6]. Antioxidant activities of most flavonoids have recently been found to be enhanced by coordination of metal ions as reported by us [7,8] and by other authors [9,10,11,12,13,14,15]. In some cases, the presence of metal ions have been found to reduce the antioxidant activities of flavonoids depending on solvent, pH and the nature of flavonoids and metal ions [16,17]. The detailed mechanism behind the effect of metal ions on antioxidative efficiency at the molecular level is far from well established.

Sn plates are very commonly used in packaging for vegetables, fruits and beverages [18,19]. The total for food packaging is approximately 80,000 million cans worldwide every year [20]. Sn cans may protect natural flavor and appearance of food through oxidation of Sn itself in preference to oxidative degradation of the food [20]. Sn used for food and beverage packaging may partly dissolve into the food preservatives and react with organic components like flavonoids [20]. Sn has been found to have a rich coordination chemistry with flavonoids including anthocyanin, quercetin and the more water soluble glucoside, rutin, as evidenced by the decrease in phenolic content of fruits canned in Sn [16,17,21]. Sn(II) coordination has been found to decrease the antioxidant effect of flavonoids [14,15]. However, molecular mechanism on radical scavenging activity of flavonoids decreased by Sn(II) coordination is still not clear.

Kaempferol (HKaem, Scheme 1a), 3,4′,5,7-tetrahydroxyflavone, a flavone derivative, is widely distributed in plants and plant products [22]. Antioxidant effects of HKaem have been reported to be increased by the presence of copper (II), iron (III), and zinc (II) by other authors and our recent studies [7,8,23,24,25,26]. In the present study, HKaem was selected as a typical flavonoid with two possible chelation sites (3,4 and 4,5) for metal ions, and complex formation by Sn(II) coordination to HKaem was investigated together with the complex as a radical scavenger. The effect on radical scavenging efficiency of HKaem by Sn(II) coordination is of relevance as Sn is used for packaging of food, and accordingly deserves more attention.

## 2. Results and Discussions

### 2.1. Formation and Stability of [Sn(II)(Kaem)_2_] Complex 

As seen in Figure 1a, HKaem in ethanol was found to react with SnCl_2_ slower than luteolin reacting with Cu(II) reaching equilibrium on a millisecond timescale [8]. The peak of UV-Vis absorption spectra red shifts gradually from 366 nm for HKaem to 433 nm by addition of SnCl_2_. The kinetics was measured every 30 s with a total time of 30 min as shown in Figure 1b. The increase at 366 nm and the decrease at 433 nm in absorbance both giving the same first-order rate constant of 0.12 ± 0.01 min^−1^. In contrast, apigenin (Scheme 1b) without 3-OH does not react with SnCl_2_ (Figure 1c), which indirectly supports that Kaem chelate to SnCl_2_ at 3-OH and 4-C=O excluding the possibility of Sn(II) chelating at 4-C=O and 5-OH [7,27,28]. This chelating mode is frequently found in metal-Kaem complexes like zinc(II) and ruthenium(II) [7,29,30].

UV-Vis absorption spectra of HKaem and Sn(II) in ratios ranging from 1:0.2 to 1:5 are seen in Figure 2a. Job’s-Plots of the spectra shown in Figure 2a at 430 nm against molar fractions of Sn(II), *F*_Sn(II)_, (Figure 2b) shows the stoichiometry of Kaem with Sn(II) is 2:1. The composition of the complex in ethanol is proposed to be a five-coordinate structure [Sn(II)(Kaem)_2_(EtOH)] containing a solvent ligand as shown in Scheme 1c, which is supported by mass spectrometry in methanol [23,31] (Table 1 and Figure 3). For simplicity, five-coordinate structure Sn(II)(Kaem)_2_(EtOH) is written as [Sn(II)(Kaem)_2_] in the following. The reaction of Sn(II) with HKaem in ethanol is thus written as Equation (1):(1)$${\mathrm{Sn}}^{2+}+2\mathrm{HKaem}\;+\mathrm{nEtOH}⇌\mathrm{Sn}\left(\mathrm{II}\right){\left(\mathrm{Kaem}\right)}_{2\;}{\left(\mathrm{EtOH}\right)}_{n}+2H^{+}$$

The stability constant was calculated as 8.2 × 10^10^ L^2^∙mol^−2^, which is closed to the stability constants, 1.1 × 10^11^ L^2^ mol^−2^ for 1:2 Cu(II)–genistein complex [23] and 1.1 × 10^11^ L^2^ mol^−2^ for 1:2 Zn(II)–kaempferol complex [7].

The stability of the [Sn(II)(Kaem)_2_] complex was investigated by addition of water, HCl and NaOH, and the corresponding absorption spectra are shown in Figure 4a–c. Figure 4a,b shows that the complex gradually decomposes with increasing addition of water or hydrochloric acid. Water may react with Sn(II) in complex to form Sn(OH)_2_ causing the dissociation of the complex as shown in Equation (2) [28,32]. The chemical equilibrium of the reaction shown in Equation (1) moves backwards by the addition of hydrochloric acid to release the parent HKaem from the [Sn(II)(Kaem)_2_] complex.
(2)$$\mathrm{Sn}\left(\mathrm{II}\right){\left(\mathrm{Kaem}\right)}_{2}+H_{2}O⇌\;\mathrm{Sn}{\left(\mathrm{OH}\right)}_{2}+2\mathrm{HKaem}$$
(3)$$\mathrm{Sn}\left(\mathrm{II}\right){\left(\mathrm{Kaem}\right)}_{2}+2H^{+}⇌{\;\mathrm{Sn}}^{2+}+2\mathrm{HKaem}$$

In addition, the [Sn(II)(Kaem)_2_] complex was also found to dissociate into deprotonated Kaem in the presence of excessive sodium hydroxide in ethanol in the reaction shown in Equation (4). The dissociation was confirmed spectrally by the similar absorption spectra of [Sn(II)(Kaem)_2_] and the parent HKaem in basic condition as seen in Figure 4c.
(4)$$\mathrm{Sn}\left(\mathrm{II}\right){\left(\mathrm{Kaem}\right)}_{2}+2{\mathrm{OH}}^{-}⇌\;\mathrm{Sn}{\left(\mathrm{OH}\right)}_{2}+2{\mathrm{Kaem}}^{-}$$

### 2.2. Radical Scavenging Kinetics 

The reaction between the semi-stable radical 2,2-diphenyl-1-picrylhydrazyl (DPPH^•^) and HKaem as well as the [Sn(II)(Kaem)_2_] complex was investigated by the absorbance changes at 517 nm in ethanol by stopped-flow spectroscopy, as shown in Figure 5.

Upon addition of HKaem only with concentration increasing from 25 to 200 μM, the decay of absorbance at 517 nm for 100 μM DPPH^•^ gradually accelerated (Figure 5a). However, the rate of DPPH^•^ scavenging gradually decreased for 100 μM HKaem with increasing addition of SnCl_2_ in ratios of HKaem:SnCl_2_ varying from 1:0.02 to 1:10 (Figure 5b). Notably, this pattern is different from what was observed in our previous and recent work including the combination of HKaem and Zn(II) or HKaem and alkaline rare earth ions in ethanol, and the combination of luteolin and Cu(II) in aqueous solution for which the rate of scavenging increased [7,8,23]. According to the method in reference [33], time evolution curves at 517 nm in Figure 5 can be fitted by the use of linear and exponential functions$,\;A_{517}=k_{1}t+b$ and $A_{517}=me^{-k_{2}t}+n$ for linear and non-linear kinetics respectively, and the initial rates at t=0 s, $rate_{t=0}$, are quantitatively obtained by differentiating the fitting functions with respect to time *t* using Equations (5) and (6):(5)$$rate_{t=0}=-\frac{dA_{517,\;t=0}}{dt}=k_{1}t+b\;\left(t=0\right)=k_{1}\;\;\;\;\;\;\;\;\;\;\;\;\;\;\;\;\;\;\;\;\;\;\;$$
(6)$$rate_{t=0}=-\frac{dA_{517,\;t=0}}{dt}=me^{-k_{2}t}\left(t=0\right)=mk_{2}\;\;\;\;\;\;\;\;\;\;\;\;\;\;\;\;\;\;\;\;\;\;\;$$
in which *k*_1_ and *k*_2_ represent rate constants and *b, m, n* are constants. The initial rates *k*_1_ for linear fitting and *k*_2_ for exponential fitting are listed in Table 2 for comparison.

The rate of DPPH^•^ scavenging dramatically decreased (~10 times) by addition of SnCl_2_ with the ratio of HKaem:SnCl_2_ changing within a small range from 1:0.02 to 1:0.08. For the ratio of HKaem:SnCl_2_ changing from 1:1 to 1:10, the rate of DPPH^•^ scavenging only decreased 2.5 times. The decrease in the DPPH^•^ scavenging rate was obviously not only the result of an increase in the fraction of the [Sn(II)(Kaem)_2_] complex.

Coordination of HKaem with Sn(OAc)_2_ as a salt of weak acid was compared with SnCl_2_ as a salt of a strong acid. The Job’s plot and stability constant were not directly available due to the poor solubility of Sn(OAc)_2_. The absorption spectra of 50 µM HKaem and 50 µM Sn(OAc)_2_ shown in Figure 6 produced a peak at 433 nm in agreement with the spectra of the [Sn(II)(Kaem)_2_] complex formed by SnCl_2_ reacting with HKaem. Absorption spectra were nicely fitted by a linear addition of the absorption spectra of HKaem and the [Sn(II)(Kaem)_2_] complex formed from HKaem reacting with SnCl_2_, which showed that the same [Sn(II)(Kaem)_2_] complex formed from both SnCl_2_ and Sn(OAc)_2_ reacting with HKaem. The stability constant for [Sn(II)(Kaem)_2_] formed from HKaem and Sn(OAc)_2_ (Equation (7)) was calculated as 5.39 × 10^8^ mol^−2^L^2^, lower than the 8.19 × 10^10^ mol^−2^L^2^ obtained for the HKaem and SnCl_2_ combination.
(7)$$\mathrm{Sn}{\left(\mathrm{OAc}\right)}_{2}+2\mathrm{HKaem}\;⇌\mathrm{Sn}\left(\mathrm{II}\right){\left(\mathrm{Kaem}\right)}_{2\;}+2\mathrm{HOAc}$$

The time evolutions of DPPH^•^ scavenging by the combination of HKaem and Sn(OAc)_2_ in Figure 5c were different from the combination of HKaem and SnCl_2_, and the radical scavenging rate of HKaem slightly decreased by Sn(OAc)_2_ coordination compared with SnCl_2_ coordination. Acetic acid, as a product formed by the reaction of HKaem and Sn(OAc)_2_, has an acid dissociation constant, *pK*_a_ = 10.59 [34], and the protons in solution were far less completely dissociated than the hydrochloric acid formed in reaction of HKaem and SnCl_2_. Therefore, the acidic effect on DPPH^•^ scavenging was negligible for the complex formed from HKaem and Sn(OAc)_2_. The [Sn(II)(Kaem)_2_] complex was the dominant DPPH^•^ radical scavenger in a solution of HKaem and Sn(OAc)_2_ excluding the interference of acid. The dramatically decreased antioxidant effect of the Sn(II)(Kaem)_2_ complex formed in the reaction with SnCl_2_ and HKaem arose from hydrochloric acid as a side product, which decreased radical scavenging activity of [Sn(II)(Kaem)_2_]. As shown in Equations (8)–(10), the mechanism of DPPH^•^ radical scavenging by [Sn(II)(Kaem)_2_] can accordingly be described as a sequential proton loss electron transfer (SPLET) mechanism [35]:(8)$$\mathrm{Sn}\left(\mathrm{II}\right){\left(\mathrm{Kaem}\right)}_{2}\to \mathrm{Sn}\left(\mathrm{II}\right){\left(\mathrm{Kaem}\right)}_{2}{}^{-}+H^{+}$$
(9)$$\mathrm{Sn}\left(\mathrm{II}\right){\left(\mathrm{Kaem}\right)}_{2}{}^{-}+{\mathrm{DPPH}}^{}\to \mathrm{Sn}\left(\mathrm{II}\right){\left(\mathrm{Kaem}\right)}_{2}{}^{}+{\mathrm{DPPH}}^{-}$$
(10)$${\;\mathrm{DPPH}}^{-}+H^{+}\to \mathrm{DPPH}H$$

Kaem, as a ligand in the complex, deprotonates first and then reacts with DPPH^•^. The deprotonated flavonoids have higher radical scavenging capacity than the protonated flavonoids. The hydrochloric acid formed from SnCl_2_ reacting with HKaem inhibited the deprotonation of HKaem in the complex and accordingly decreased the radical scavenging reactivity of the [Sn(II)(Kaem)_2_] complex. The addition of hydrochloric acid to the solution of HKaem and Sn(OAc)_2_ significantly decreased the rate of DPPH^•^ decay, as seen in Figure 5c, which further indicated the inhibition mechanism of acid on radical scavenging of the [Sn(II)(Kaem)_2_] complex.

The absorption spectra of DPPH^•^ radicals with HKaem, SnCl_2_/Sn(OAc)_2_ and equilibrated solutions of HKaem and SnCl_2_/Sn(OAc)_2_ shown in Figure 7 also indicated that Sn(II) salts alone do not react with DPPH^•^. HKaem and the combination of Kaem and Sn(OAc)_2_ were able to scavenge the DPPH^•^ radicals. The spectral characteristics of DPPH^•^ are not affected by combination of HKaem and SnCl_2_, which excludes the possibility of DPPH^•^ protonation in the radical scavenging [36,37].

The time evolution of absorption of 50 μM DPPH^•^ at 517 nm following addition of 100 μM HKaem soaked in a glass vessel and in a Sn can for 48 h (Figure 8) indicated that the radical scavenging rate of HKaem is 0.72 times lower than of the parent HKaem and is apparently decreased by reaction with Sn as the packaging material in cans. This result implied that Sn or Sn oxide may dissolve and react with HKaem in solution forming a complex, which causes a decrease in radical scavenging efficiency.

Decreased radical scavenging reactivity of HKaem by Sn(II) coordination is also supported by comparisons of oxidation potential as determined for Kaem and [Sn(II)(Kaem)_2_] formed from Kaem reacting with SnCl_2_ and Sn(OAc)_2_ (Figure 9a,b). Using cyclic voltammetry, HKaem was found to be oxidized by a quasi-reversible process and to have an oxidation potential of 0.08 V versus ferrocene corresponding to oxidation of a phenolic group [38,39]. No signal was observed for SnCl_2_ or Sn(OAc)_2_ alone within the detection range. The lowest oxidation peak of HKaem gradually increased with the addition of SnCl_2_ with the ratio of HKaem:SnCl_2_ at 1:0.1 to 1:5. With the increase in SnCl_2_ relative to HKaem, a new oxidation peak appeared and moved toward a higher oxidation potential of 0.41 V, assigned to the oxidation potential of the [Sn(II)(Kaem)_2_] complex formed from HKaem reacting with SnCl_2_ (Figure 9a). The oxidation potential of the Sn(II)(Kaem)_2_ complex formed from HKaem reacting with Sn(OAc)_2_ was determined to be 0.16 V (Figure 9b). This value is higher than the potential of HKaem of 0.08 V but lower than the potential of the [Sn(II)(Kaem)_2_] complex formed from combination of Kaem and SnCl_2_, which had a value of 0.41 V. A higher oxidation potential for HKaem coordination to SnCl_2_ than for HKaem coordination to Sn(OAc)_2_ confirmed that the SPLET reaction of the [Sn(II)(Kaem)_2_] complex in radical scavenging also occurs at the electrode during cyclic voltammetry. The higher oxidation potential of the [Sn(II)(Kaem)_2_] complex than the parent HKaem is consistent with the decreased radical scavenging capacity of HKaem by Sn(II) coordination.

As a main group metal element, Sn(II) often has metal–ligand covalent interactions with other elements [40]. Cationic low-valent Sn(II) species possess a combination of electrophilicity of the cation with the nucleophilicity of the metal-centered lone electron pair [41]. Compared with their parent flavonoid molecules, Sn(II) reacting with HKaem decreases the radical scavenging efficiency and increases oxidation potential, whereas transition metal ions like Zn(II) and Cu(II) reacting with flavonoids both increase the radical scavenging efficiencies and decrease the oxidation potentials. This may arise from the weakened electron donation ability of HKaem by coordination of Sn(II) with the nucleophilicity of the metal-centered lone electron pair, whereas the strong electron-withdrawing effects of Zn(II) or Cu(II) increase the electron donation ability of flavonoids in radical scavenging.

## 3. Materials and Methods

### 3.1. Chemicals

HKaem (>98%) and HApi (>98%) from Huike Plant Exploitation Inc, (Shanxi, China), ‘stannous chloride dihydrate, SnCl_2_·2H_2_O (>99%) from Beijing Chemical Reagents Company (China), tin acetate (Sn(OAc)_2_, C_4_H_6_O_4_Sn, > 97%) from Energy Chemical (Shanghai, China), DPPH^•^ (>97%) from Zhongshengruitai Technology Inc. (Beijing, China), ferrocene (>98%) and sodium perchlorate, NaClO_4_ (>98%) from Sigma-Aldrich (St. Louis, MO, USA), hydrochloric acid (37%) from Sigma-Aldrich, NaOH (>98%) from Sigma-Aldrich, and spectrophotometric grade ethanol and methanol (99.9%, Fine Chemical Industry Research Institute, Tianjin, China) were used as received. Ultrapure water purified on a Milli-Q purification train was used throughout. Sn cans used to keep fruits were bought from supermarkets and were polished to remove the oxide layer before performing the experiment.

### 3.2. Reaction of Sn(II) with Kaempferol

All UV-Vis absorption spectra were measured on a Cary 60 spectrophotometer (Varian, Inc., Palo Alto, CA, USA) using 1.0 cm quartz cells in a room at a temperature of 25 °C. The solutions were prepared by mixing solutions of HKaem and SnCl_2_ with total molar concentrations of 50 μM in Kaem:Sn(II) molar ratios varying from 9:1 to 1:9. Stabilities of the complex were investigated by addition of different concentrations of water, acid, and base to the solution of 50 µM SnCl_2_ and 50 µM HKaem. For the reaction of HKaem with Sn(OAc)_2_, 100 μM HKaem was added to 100 and 200 μM Sn(OAc)_2_ in ethanol. The absorption spectra to monitor the reactions were measured after 30 min when the reactions had reached equilibrium. For the reaction of HApi with SnCl_2_, 100 µM HApi was added to 100 µM SnCl_2_ in ethanol and stored for 30 min and 24 h. Then, UV-Vis spectroscopy was performed for the two solutions.

### 3.3. Mass Spectroscopy

Mass spectra were obtained on a Thermo Scientific™ Q Exactive™ HF (Waltham, MA, USA) in positive ion mode. The [Sn(II)(Kaem)_2_] complexes were prepared by filtering the solutions obtained by mixing 100 µM HKaem + 50 µM SnCl_2_ through a nylon membrane with 220 nm sieve pores. The samples were analyzed by direct infusion into electron spray ionization by means of a syringe pump (Thermo UltiMate 3000, Waltham, MA, USA) at a flow rate of 5 μL/min. Capillary temperature was 320 °C and spray voltage was 3.50 kV.

### 3.4. DPPH^•^ Radical Scavenging

The kinetics of DPPH^•^ scavenging by the [Sn(II)(Kaem)_2_] complexes was investigated using the same rapid mixing stopped-flow technique, performed on a RX2000 Rapid-Mixing Stopped-Flow Unit (Applied Photophysics Ltd., Surrey, U.K.) as in our previous study [7]. One syringe contained a solution of DPPH^•^ dissolved in ethanol. The other syringe contained the samples to be measured.

The kinetics of DPPH^•^ scavenging by a HKaem in ethanol soaked in a Sn can for 48 h was compared to a similar sample under the same conditions but stored in a glass vessel covered by aluminium foil.

### 3.5. Determination of Oxidation Potentials

Cyclic voltammetry (CV) was performed on a three-electrode CHI 760D electrochemical analyzer (ChenHua Instruments Inc., Shanghai, China). The working electrode was a glassy carbon piece (diameter = 4 mm), the reference electrode was a silver wire and the auxiliary electrode was a platinum wire. The supporting electrolyte, 0.10 mol L^−1^ NaClO_4_, was used. The internal standard, 5.0 × 10^−5^ mol L^−1^ ferrocene, was used. The cyclic voltammetry in ethanol was scanned in potential from −0.5 to 1.0 V on a 0.1 V/s scan rate.

All experimental results were repeated three times and showed the same tendency.

## 4. Conclusions

Sn(II) reacts with HKaem forming the proposed [Sn(II)(Kaem)_2_] complex by coordination at the 3,4-site on a time scale of seconds, as confirmed by UV-Vis spectroscopy, mass spectroscopy, and through a comparison with reaction of Sn(II) and apigenin. Radical scavenging reactivity of HKaem was found to be moderately decreased by Sn(II) coordination by comparing the complexes formed from SnCl_2_ and Sn(OAc)_2_ reacting with HKaem, in agreement with the increase in oxidation potential for the [Sn(II)(Kaem)_2_] complex compared to the parent HKaem, which may be due to the electron donation ability of Kaem as a ligand in the complex decreased by the nucleophilicity of the Sn(II)-centered lone electron pair. These results concerning the molecular mechanism behind the radical scavenging efficiency of HKaem as decreased by Sn(II) is relevant for Sn as food packaging in the food industry and protection of flavonoids as natural antioxidants, which needs more attention.

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
