# Peer review of "Kinetic Studies on Radical Scavenging Activity of Kaempferol Decreased by Sn(II) Binding"

_molecules, 2020, doi:10.3390/molecules25081975_

Round 1

Reviewer 1 Report

The manuscript is dealing with very interesting and novel data. Since the study was well designed and described, I suggest only some minor corrections in order to improve this manuscript.

INTRODUCTION

The following sentences are very general and thus, they are unnecessary “The introduction should briefly place the study in a broad context and highlight why it is 25 important. Flavonoids are a group of natural products with a variety of biological activities such as 26 antioxidant, anti-inflammatory and anticancer[1-3].” Please remove them and start the Introduction with the sentence “Antioxidant activities of most…”

3.4. DPPH• Radical Scavenging

Please provide proportions of the reaction mixture. How much was the Kaem added to the DPPH solution? What was the volume of the measured sample?

Reviewer 2 Report

Summary:

Han et al. report on the reaction of SnCl2 with kaempferol (HKaem, 3,5,7-Trihydroxy-2-(4-hydroxyphenyl)-4H-1-benzopyran-4-on) presumably forming a 1:2 [Sn(II)(Kaem)2] complex in ethanol solution. DPPH• scavenging efficiency of Kaem is dramatically decreased by addition of SnCl2, supporting the binding of Sn(II) to Kaem. Sn-binding assumed at the 3,4 positions of Kaem inhibits deprotonation of HKaem thus reducing the radical scavenging activity of complex via a sequential proton-loss electron transfer (SPLET) mechanism. Moderate decreases in radical scavenging for HKaem are observed by adding Sn(OAc)2 and by storing an ethanol solution of HKaem in tin cans, which is in line with an increase of oxidation potential from kaempferol to the complex and may be related to the decrease of nutritive value for fruits and vegetables with tin as package materials.

Evaluation:

In continuation of their recent work the authors present a study in which SnCl2 and Sn(OAc)2 were mixed with the interesting flavone HKaem (3,5,7-Trihydroxy-2-(4-hydroxyphenyl)-4H-1-benzopyran-4-on). The resulting solution were studied for their spectroscopic, electrochemical and DPPH• scavenging properties.

In the present state the manuscript is not publishable in Molecules, the quality of this report is far too small. Reasons are:

  1. The most critical point is that the proposed Sn(II) complex was not isolated and characterised. The UV spectroscopy and MS point to such a complex, but these methods are not completely unequivocal, nor do they allow to draw a precise molecular structure as done in Scheme 1. Moreover, if the complex [Sn(Kaem)2] is responsible for the observed phenomenae, why is there a difference between the use of SnCl2 and Sn(OAc)2? So, the authors need to isolate and characterise the supposed complex. This cannot be difficult for chemists. 1H, 13 and 119Sn NMR spectroscopy will be great tools to determine, in combination with elemental analysis, exactly the complex structure.
  2. Essential references were missing. Especially the studies by De Giovani et al from 2004 and 2005 are important for the discussion of all the obtained results (see list below).
  3. Since Molecules is a journal for generalists, the concepts, the nomenclature and formats must be precise and correct. However, presentation and writing are both very poor in this manuscript, confusing largely the reader. The English is bad and must be revised by somebody of fluent English, ideally a native English speaker. The formulae and nomenclature must be revised. Typoos must be removed.
  4. Finally, numerous studies on the radical scavenging activity of flavones in the absence of presence of metal ions have been reported. This report just adding one more with the same conclusion as the other. A slightly new twist is the use of Sn(II) instead of transition metals. Thus, this report is of only low overall importance.

Without adding more experimental support for the claims and massive editing of the manuscript, this report is not publishable. Therefore, I suggest to reject the manuscript. After adding more experiments, the authors might re-submit a revised manuscript.

Missing references that must be included in the introduction and/or discussion:

  1. De Giovani et al. RedoxRep 2004 Antioxidant properties of complexes of flavonoids with metal ions.
  2. De Giovani et al. SpectrochimActaA 2005 Synthesis, spectral and electrochemical properties of Al(III) and Zn(II) complexes with flavonoids.
  3. Raman et al. ApplOrganometChem 2018 mixed ligand complexes using flavonoids as precursors Co Ni Cu Zn.
  4. Woznicka et al. JInorgBiochem 2014 Syntheses, crystal structures and antioxidant study of Zn(II) complexes with morin-5'-sulfonic acid (MSA).
  5. Estevinho et al. Molecules 2019 Kaempferol A Key Emphasis to Its Anti-cancer prop review.
  6. Szerb et al. Polyhedron 2018 New heteroleptic Zn(II) and Cu(II) complexes with quercetine and N ˆN ligands.
  7. Guo et al. BiolTraceElemRes 2014 Zinc-Binding Sites on Selected Flavonoids review.
  8. Guo et al. Molecules 2015 Studies Fe, Co, Ni, Cu and Zn Quercetin Complexes ESI-MS
  9. Lewandowski et al. JThermAnalCalor 2016 Spectroscopic, thermogravimetric and biological studies of Na(I), Ni(II) and Zn(II) complexes of quercetin.

Points for a revision:

  1. Abstract, the name of kaempherol: 3,5,7-Trihydroxy-2-(4-hydroxyphenyl)-4H-1-benzopyran-4-on should be mentioned here and in the introduction alongside with the information that this is a flavone derivative.
  2. Abstract, line 14 and further through the manuscript. The assumed complex should be called [Sn(Kaem)2] or [Sn(II)(Kaem)2] using the established complex nomenclature. The “1:2” can be deleted.
  3. Abstract and further through the manuscript. The authors have to make clear that HKaem is the 3-hydroxy-flavone molecule. When binding it gets deprotonated and should thus be called Kaem. The used term “Kaem-H” is misleading.
  4. Abstract and further through the manuscript. The complex [Sn(Kaem)2] has not been isolated and characterised. Therefore, this structure is a claim or suggestion. This must be made clear. Also the binding to the 3,4-site of Kaem- is assumed. It is reasonable to assume that Sn2+ ions bind to this site, since many complexes show this binding motive. Still, without a molecular structure from single crystal XRD, this remains a claim. So, the first sentence should be rewritten as shown in the Summary above. All over the manuscript, the authors must be clear in this point.
  5. Introduction, delete the first sentence.
  6. Scheme 1 … (c) proposed structure of the [Sn(Keam)2] complex
  7. Scheme 1, delete the charges
  8. Page 2, and all over the manuscript. Oxidation states are immediately following the element or its abbreviation: copper(II), Cu(II), Sn(II) etc.
  9. Page 2, last paragraph. We do NOT see from Figure 1a that Kaem binds slower to SnCl2 than other metals. The reaction rates must be provided here and a comparison cannot be drawn from the Figure in the very first sentence.
  10. Page 2, line 59, what do you mean by “stable structure of SnCl2”. First of all, SnCl2 is a solid. SnCl2 in ethanol solution contains probably species such as [SnCl2(HOEt)x] with x something 2 – 5.
  11. Page 2, line 62. There are numerous crystal structure showing metals binding to Kaem- in this way. Proper discussion and citing of references is missing here.
  12. Page 3, equation (1) and all over the manuscript. The protonated molecule should be called HKaem, the deprotonated Kaem-.
  13. Page 4, why is the MS study carried out in MeOH and not in EtOH?
  14. Page 5, just one example of the poor English: “The stability of the Sn(II)-Kaem2 complex was performed by addition of water, hydrochloric acid …”
  15. Page 7, Figure 6, which solvent?
  16. Page 8, Figure 7, which solvent? The same for Figure 8.
  17. Page 10, Figure 9, which solvent?
  18. Page 10, last paragraph. What about details of the reaction of HKaem with Sn(OAc)2 and the reaction of SnCl2 with api? I could not find them.

Reviewer 3 Report

Manuscript of Han and co-workers describes the chelation effect of Tin2+ ion to kaempferol, in particular focusing the attention on the scavenging effect. The 2:1 complex between kaempferol and Tin ion was demonstrated by UV-Vis titration, Job’s Plot and mass spectrometry. In addition, extended kinetic studies on radical scavenging were reported.

The manuscript is well written, the experiments well designed and performed, thus I suggest the publication on Molecules after some minor revisions:

  • In the introduction section, the first row is probably wrong, please check;
  • The authors calculated the stability constant between kaempferol and Tin, but with equation or software? In addition, due to the slow kinetic of the complexation equilibrium, the UV spectra used to obtain the binding constant were measured after what time? These points should be clarify;
  • Some compounds in the manuscript are not correctly reported: “acetate acid”, “hydrochloric acid”.

Reviewer 4 Report

The paper entitled “Kinetic studies on radical scavenging activity of 2 kaempferol decreased by Tin (II) binding” describes the formation of Sn-kampeferol complex and the effect of the formation of this complex on the radical scavenging activity of kampeferol when complexed with Sn. This study is interesting as it describes the ability of kampeferol present in various fruits and vegetables with Sn that is present in cans used for producing canned fruits and vegetables. Nevertheless taking into account that the presence of water dissociates the complex that is synthesized in ethanol, and that fruits and vegetables are canned with water solution, what is the opinion of the authors for the meaningfulness of these reaction when the fruits and vegetables are canned. Also as low pH also acts in dissociation of the complex, and most of the fruits are canned in acidic solutions what do you think it will happen to the complex?

One point that is not clear in material and methods and also in the results and discussion section, is when the kampeferol was soaked in the Sn can, it was in water or ethanol?

Although the study was carefully planned and the methods used allow to understand the mechanism of complex formation and also its effect on the antioxidant activity of kampeferol what is the real importance and application of this reaction for canned foods or other?

In the introduction the authors didn’t remove the first sentence that is present by default in the template

 The introduction should briefly place the study in a broad context and highlight why it is 25 important. Flavonoids are a group of natural products with a variety of biological activities such as 26

Please explain better this sentence in light of the results obtain in the work

and the effect of the complex as a radical scavenger. Decreased radical scavenging efficiency of Kaem 47 by tin (II) coordination may moreover related to tin used for package of food, which deserved more 48 attention. 49

There is no acetate acid, there is acetic acid. Nevertheless the pKa described is for aqueous solution and the complexation is performed in ethanol, therefore the dissociation of acetic acid as well hydrochloride acid are affected by ethanol. This should be better explained using the acids dissociation constants in ethanol.

Acetate acid as a product formed in reaction of Kaem and SnAc2 has a pKa = 4.75[25] 153 and the protons in solution is far less than completely dissociated hydrochloric acid formed in reaction of Kaem and SnCl2. 154

From this sentence not only the Sn-Kamp complex affect Kamp antioxidant activity but as well the presence of HCl formed by formation of Kamp-Sn complex when using the SnCl2 salt. Why not to neutralize the HCl formed by using a base like an ethanol soluble amine and found more evidence for this sentence?

Sn(II)- Kaem2 complex is the dominate DPPH• radical scavenger in solution of Kaem and SnAc2 excluding 156 the interference of acid. The dramatically decreased antioxidant effect of Sn(II)-Kaem2 complex 157 formed in the reaction with SnCl2 and Kaem arises from the hydrochloric acid as a side product, 158 which decreased radical scavenging activity of Sn(II)-Kaem2. 159

Conclusion: please explain better this sentence based on your results

Present results about molecular 261 mechanism on radical scavenging efficiency of Kaem decreased by Sn(II) may be related to tin as food 262 package in food industry and protection of flavonoids as natural antioxidants draws more attention

Phenolic compounds are not nutritive components of foods because they have no nutritional value. Please explain better what you want to say.

Abstract

 Moderate decreases in radical scavenging for Kaem are observed by Sn(CH3COO)2 coordination 18 and by storing in tin cans, which is supported by the increase of oxidation potential from kaempferol 19 to complex, and may be related to the decrease of nutritive value for fruits and vegetables with tin 20 as package materials.

Round 2

Reviewer 2 Report

The authors have revised their manuscript according to a number of reviewers comments. However, for the most important point, the characterisation of the new compound [Sn(Kaem)2] they failed.

In the guidelines of Molecules it is clearly outlined that …”Complete characterization data must be given for all new compounds” …

The evidence for this complex comes from a Job plot which is in line with a 1:2 ratio of metal and Kaem and MS. The mass spectrometry in Table 1 shows species consisting of Sn(II) 2 Kaem ligands and MeOH. From MS it seems that MeOH is another ligand and the structure would be thus [Sn(Kaem)2(MeOH)]. But after all, MS is not able to show this unequivocally.

The authors write: …”The structure of Sn(Kaem)2 is confirmed in the present study based on Job-plot method, mass spectra, spectral comparisons with the data of apigenin as ligand as well as the spectra of other metal-kaempferol complexes reported in literature and in our previous study” …

I must insist, the structure is NOT CONFIRMED.

As long as the authors still insist that they “confirmed” this complex, I see no way to publish this material in a chemical (!) journal such as Molecules.

As I can understand that under the present circumstances further experiments are not possible or might consume too much time, I can offer the following option: If the author remove all their unjustified claims that they “confirmed” the structure of the complex and replace it by appropriate careful descriptions in a detailed revision, I can think about acceptance.

This means that the composition and structure of the complex are “proposed structures” or “assumed structures” which were “supported” (not confirmed) by the Job plot, the UV-vis and the MS experiments. The description and discussion must be led on “proposed complex species”, the “assumed [Sn(Kaem)2(EtOH)n] complex” and so on.

From a chemical viewpoint it is also clear, that in solution, there is probably no such complex [Sn(Kaem)2]. Sn2+ is a pretty big ion and its coordination numbers (number of ligands) in solution lie around 5-7. Thus the structures depicted in Scheme 1 are probably not representing the complex in solution. Strong indication for this comes from MS (see later).

Points for a revision:

  1. I must insist on the much more versatile use of HKaem as the 3-hydroxy-flavone molecule and Kaem for the deprotonated ligand. The used term “Kaem-H” is highly misleading and also extremely cumbersome.
  2. Please replace tin by Sn everywhere
  3. Abstract: Kaempherol (HKaem = …) was admixed to SnCl2 and Sn(OAc)2 in ethanol and the DPPH scavenging activity of these solutions was studied. In the presence of SnCl2 the scavenging activity of HKaem is dramatically decrease, presumably through Sn coordination. A Job-plot from UV-vis absorption spectroscopy when mixing SnCl2 and HKaem shows a 1:2 ratio for the formed complex. MS spectroscopy of MeOH solutions confirms the 1:2 ratio but for all species MeOH seems to be a further ligand. The assumed complex species in ethanol are thus [Sn(Kaem)2(EtOH)n] (n = 1 or 2) containing two deprotonated kaempherol ligands. The formation of these complexes yields H+ and thus reduces the radical scavenging activity via … mechanism. Upon addition of Sn(OAc)2 only a moderate decrease of activity is observed, in line with the poorer solubility of Sn(OAc)2 and the competing coordination of acetate, preventing gross formation of the Kaem complex. Cyclic voltammetry showed an increase in the oxidation potentials upon addition of SnCl2 to HKaem in line with complex formation. This altered redox behaviour may be related … materials.

(comment: I hope this part makes the requested careful writing clear).

  1. Scheme 1, write: Molecular structures of (a) HKaem, (b) Api and (c) proposed structure of the complex [Sn(Kaem)2(EtOH)]
  2. Please draw this five-coordinate species – four-coordinate Sn(II) is very unlike.
  3. Line 60, Formation and stability of Sn(II) kaempherol complexes
  4. Line 61. As seen in Figure 1a binding of Kaem to Sn(II) in SnCl2-containing ethanol occurs slower than luteolin binding by Cu(II) … (references needed)
  5. Line 67, delete the sentence: The slower … as ligand.
  6. Line 68 and further in the manuscript. Your idea that SnCl2 binds to Kaem is very probably confusing the reader and its not true. From your study it is clear, that if Kaem binds to Sn(II) the chlorides will be release. If you think about this process step-by step, a complex species like [Sn((Kaem)Cl] might play a role, this would be a 1:1 complex. But your job plot clearly shows that a 1:2 complex is far more stable than a 1:1 complex. So forget this idea SnCl2 binding to Kaem and rewrite all paragraphs accordingly
  7. Line 71. This binding mode is frequently found in Kaem complexes.(references)
  8. Line 74. Figure 1. Slow reaction of HKaem and no reaction of HApi in solutions of SnCl2…
  9. Line 83, equation (1): Sn2+ + 2HKaem = [Sn(Kaem)2(EtOH)n] + H+ (comment: here you can clearly see that my recommendation of point 1 is very helpful. Your equation is highly confusing)
  10. Line 85, Figues 2. Determination of the Sn:Kaem ratio. (a) …
  11. From the MS experiments I would conclude: … the MS experiments is in line with the Sn:Kaem 1:2 ratio of the Job plots (Figure 2). However, MeOH seems to be also a ligand in the Sn(Kaem) complex species observed in the MS experiments. Thus, we assume that in ethanol solution five or six-coordinate complexes [Sn(Kaem)2(EtOH)n] (n = 1 or 2) are predominant.
  12. Electrochemistry. The conclusion from these measurements is clear. You observe the oxidation of the HKaem ligand in pure HKaem solutions. In Sn-containing solution Sn complexes were formed and HKaem is deprotonated. The Sn-bound Kaem- anions have higher oxidation potentials due to their negative charge.
  13. Line 234, comment to: "In contrast …" Here, you propose that the [Sn(Kaem)2] complex is different when formed from SnCl2 or from Sn(OAc)2. This is confusing and probably wrong, see point 9. But maybe the oberseved species are different. Using a 1:1 molar ratio of HKaem and SnCl2 or Sn(OAc)2 probably clearly yields [Sn(Kaem)2] from SnCl2 solutions. But in Sn(OAc)2 solutions, the 1:1 species [Sn(Kaem)(OAc)] might play a marked role besides [Sn(Kaem)2]
  14. Line 250. Delete “post transition”. Sn is a “main group metal”. The entire sentence is confusing. Main group metal ions as Sn2+ have flexible coordination numbers and no preferences to high geometry polyhedra in their complexes. Metal-ligand bonds range from polar covalent to highly ionic. Cationic …
  15. Line 318, … Sn(II) binds to deprotonated Kaem ((HKaem, 3,5,7-Trihydroxy-2-(4-hydroxyphenyl)-4H-1-benzopyran-4-on) presumably at the 3,4 site on the time scale of … apigenin. Job-plots based on UV-vis absorption spectroscopy of a reaction mixture of SnCl2 and HKaem gave a 1:2 ratio for the formed complex, while MS on such solution gave evidence for further ligands. Thus, in ethanol solutions species like [Sn(Kaem)2(EtOH)n (n = 1 or 2) were assumed. Radical scavenging activities of solutions containing HKaem and SnCl2 or Sn(OAc)2 are decreased compared with solution of pure HKaem and a sequential proton-loss electron transfer (SPLET) mechanism is supposed since the coordination of HKaem yields protons. The oxidation potentials of HKaem are also markedly altered in the presence of SnCl2 or Sn(OAc)2. Formation of Sn(II) Kaem complexes obviously leads to increased oxidation potentials for the Kaem-centered oxidations, which has an implication for the use of tin for food packaging. Quite generally, protection of … antioxidants from unwanted interaction with the packaging material clearly … attention.

This is not an extensive list. I expect that the authors revise their manuscript word-by-word not only the 15 points of this list.

Reviewer 4 Report

Dear authors and editors, although the authors performed a subtancial change in the manuscript, I don't believe that they have answered all the questions made in the previous review that are in attachment.

Also in the revised review please correct

Kaempferol (Kaem, Scheme 1a), 3,5,7-trihydroxyflavone, is a common type of flavone

48 derivative, is widely distributed in plants and plant-derives products

Kampeferol is a flavonol and it is 3,4′,5,7-Tetrahydroxyflavone and not 3,5,7-trihydroxyflavone

Kind regards
